# Clinical Outcomes in Older Patients Aged over 75 Years Who Underwent Early Surgical Treatment for Pyogenic Vertebral Osteomyelitis

**DOI:** 10.3390/jcm10225451

**Published:** 2021-11-22

**Authors:** Jeong Hwan Lee, Jihye Kim, Tae-Hwan Kim

**Affiliations:** 1Spine Center, Department of Orthopedics, Hallym University Sacred Heart Hospital, Hallym University College of Medicine, Gyeonggi-do, Anyang 14068, Korea; 160129@hallym.or.kr; 2Department of Pediatrics, Division of Infection, Kangdong Sacred Heart Hospital, Hallym University College of Medicine, Seoul 05355, Korea; jihyewiz17@kdh.or.kr

**Keywords:** pyogenic vertebral osteomyelitis, spondylodiscitis, neurologic deficit, spinal surgery, decompression, instrumentation, risk factor, recurrence, mortality

## Abstract

Older patients with pyogenic vertebral osteomyelitis (PVO) usually have more medical comorbidities compared with younger patients, and present with advanced infections from different causative organisms. To aid surgical decision-making, we compared surgical outcomes of older patients with PVO to those who underwent nonoperative treatment. We identified the risk factors for adverse post-operative outcomes, and analyzed the clinical risks from further spinal instrumentation. This retrospective comparative study included 439 patients aged ≥75 years with PVO. Multivariable analysis was performed to compare treatment outcomes among three groups: 194, 130, and 115 patients in the non-operative, non-instrumented, and instrumented groups, respectively. The risk factors for adverse outcomes after surgical treatment were evaluated using a logistic regression model, and the estimates of the multivariable models were internally validated using bootstrap samples. Recurrence and mortality of these patients were closely associated with neurologic deficits, and increased surgical invasiveness, resulting from additional spinal instrumentation, did not increase the risk of recurrence or mortality. We propose that surgical treatment for these patients should focus on improving neurologic deficits through immediate and sufficient removal of abscesses. Spinal instrumentation can be performed if indicated, within reasonable clinical risk.

## 1. Introduction

In recent decades, the incidence of pyogenic vertebral osteomyelitis (PVO) has increased, particularly in older patients with comorbidities [1,2,3]. This increase has been attributed to advances in diagnostic imaging technologies, increased number of spinal procedures, and increased life expectancy [4]. Although most patients with PVO present with an indolent course, the diagnosis of PVO is frequently delayed in older patients, who present with advanced infection at the time of diagnosis and often die prematurely from hemodynamic instability due to sepsis, precluding further management. In the case of survival, serious morbidities secondary to subsequent structural and neurological injury, such as paralysis and kyphotic deformity, and resultant mortality may later arise in this unique cohort [5,6].

Recent comparative studies have consistently reported that early surgical treatment using spinal instrumentation in patients with PVO has favorable clinical outcomes with respect to recurrence and mortality [7,8]. Studies have also shown favorable outcomes in PVO patients with previous spinal instrumentation and end-stage renal disease, which have been considered the most prominent risk factors for the recurrence of PVO [9,10,11]. Accelerated bone healing resulting from firm stabilization and sufficient abscess removal during surgical treatment using instrumentation may compensate for the risk of recurrent infection due to foreign body insertion [10].

Early surgical treatment can be particularly useful for older patients with PVO. Sufficient removal of abscesses increases the likelihood of recovery from preoperative neurologic impairment. A stabilized spine enables early ambulation and can prevent morbidities arising from prolonged immobilization, such as venous thromboembolism and organ system dysfunction, including cardiopulmonary, endocrine, neurologic, psychiatric, gastrointestinal, and urinary tract dysfunction [12,13]. Nevertheless, early surgical treatment using spinal instrumentation should be carefully considered in older patients with PVO, who may be at a greater risk of mortality from invasive surgical procedures and aggravation of previous comorbidities. In this respect, previous studies involving younger patients have limited applicability for planning treatment for older patients with PVO. This cohort is thought to have unique susceptibility patterns to different causative organisms, more severe comorbid medical conditions, and reduced bone quality. Therefore, the clinical outcomes of early surgical treatment for PVO should be analyzed separately in older populations considering their clinical characteristics. However, no studies have evaluated the clinical outcomes of early surgical intervention for these patients.

Our study aimed to aid clinical decision-making for older patients with PVO who require surgery due to structural instability, neurologic deficits, or uncontrolled infection. Even among older patients with PVO, most patients with a less advanced infection can easily be treated with appropriate antibiotics. Therefore, we initially compared the surgical outcomes of older patients with PVO to those who underwent nonoperative treatment. Subsequently, the risk factors for adverse clinical outcomes after surgery were evaluated to guide appropriate selection of candidates for surgery. In addition, clinical risks from additional spinal instrumentation were thoroughly analyzed by comparing adverse clinical outcomes between the instrumented and non-instrumented surgical groups. Recommendations for surgical treatment based on these analyses were accordingly proposed for this patient population.

## 2. Materials and Methods

### 2.1. Study Design and Ethics

This retrospective comparative study included patients with PVO who visited our institution between January 2004 and September 2020. Older patients aged ≥75 years who underwent medico-surgical treatment for PVO were considered eligible. PVO was diagnosed as previously described [8,9,10]. In patients with PVO, early surgical treatment generally means operative management during the initial antibiotic treatment. Considering a 6-week course of intravenous antibiotics according to the guidelines for PVO, [14] early surgical treatment was defined as operative management done during the initial 6-week treatment. We excluded patients who underwent delayed (i.e., >6 weeks from PVO diagnosis) surgery for PVO, those who had a history of previous spinal instrumentation at the same site of infection, those who had a malignancy or persistent open wounds (including pressure ulcers), and those with incomplete medical records (Figure 1). This study was designed and conducted according to the guidelines of the Declaration of Helsinki and was approved by our institutional review board (IRB number: 2018-08-016). The requirement for informed consent was accordingly waived by this authority.

### 2.2. Covariables and Treatment Outcomes

The demographic data and precise medical condition of the patients were retrieved from electronic medical records. Charlson comorbidity index (CCI) scores were subsequently calculated. The non-age-adjusted version of the CCI was used for independent estimation of the effect of age on surgical outcomes, through multivariable adjustment [15]. The degree of neurological impairment was classified according to the American Spinal Injury Association impairment grade (ASIA grade). Data on bone mineral density at the time of diagnosis of infection were obtained. The severity of infection was classified according to the system proposed by Pola et al. [16]. Data on the anatomical location of infection, including that of the primary lesion, presence of abscess, and number of infected bodies were obtained from initial MRI studies. Multiple primary regions were defined as non-contiguous, separate foci of infection, involving at least two sites. If a single, continuous focus of infection exists involving two regions, this type of infection was classified based on the region with a more severe degree of involvement. Data on culture studies and initial laboratory values, including white blood cell count (WBC), erythrocyte sedimentation rate (ESR), and C-reactive protein (CRP), were also retrieved.

Implant failure and major medical events after treatment were defined according to a previous study [7]. Recurrence was defined as recurrent signs and symptoms after completion of the antibiotic course and receiving a second course of parenteral antibiotics [9,10,17]. Recurrent symptoms were defined by the occurrence of axial pain, or pain and weakness in the extremities, which could be explained by anatomical involvement of the infection. Recurrent signs were defined as fever or neurologic deficit that was demonstrated through the ASIA grade. Mortality was accordingly classified as either 90-day or 1-year mortality.

### 2.3. Protocol for Surgical Treatment of Older Patients with PVO

Surgery was performed upon presentation of substantial or aggravating neurologic deficits, intractable pain due to abscess, or a major deformity or mechanical instability before or after abscess drainage. Surgical management in PVO patients is not intended to remove all infected structures, but to stabilize the spine and preserve bone and soft tissue structures while effectively draining the abscess, which is associated with neurologic deficit and recurrence of infection [8,9,10]. Therefore, based on preoperative MRI, an optimal surgical approach that can remove the abscesses most effectively was used. Generally, the posterior approach is preferred for simultaneous conduct of rigid fixation. If a significant amount of psoas abscess was present, percutaneous catheter drainage was performed, preoperatively.

Epidural abscesses were initially drained by performing hemilaminectomy, and the liquefied intervertebral abscesses were drained simultaneously. If abscess drainage was incomplete because of adhesion to the underlying dural structure, additional laminectomy was performed while preserving the facet joint. Spinal instrumentation was implemented if a major deformity or mechanical instability was present before or after surgical abscess drainage. Long segment fixation with a pedicle screw that sufficiently reaches the anterior cortex (bicortical), including the involved vertebral body and spanning over and below two portions of the infected segment, was preferred. Cement augmentation was generally avoided to ensure interosseous circulation, as was cage insertion into the interbody space if an intervertebral abscess was present, preoperatively. If required, interbody fusion was performed using an autologous iliac or allograft bone.

### 2.4. Statistical Analysis 1: Comparison of Treatment Outcomes According to the Three Treatment Groups in the Whole Cohort

We divided the whole cohort of patients into three groups, based on treatment approach. Initially, the patients were divided into those who underwent nonoperative management and those who underwent surgical intervention. Those who underwent surgery were further classified based on whether they underwent additional instrumentation (Figure 1). We compared baseline patient characteristics, infection profiles, and treatment outcomes according to the treatment methods. Recurrence and death-free survival at 1-year follow-up were estimated. These were presented by survival probability and corresponding 95% confidence interval using the method by Hosmer-Lemeshow. Finally, multivariable analysis was performed to evaluate the relative risks of recurrence and mortality in the surgical groups compared to the non-operative group. Adjustments were made for independent variables that showed differences (*p* < 0.05) among the three groups.

### 2.5. Statistical Analysis 2: Identification of Risk Factors for Adverse Outcomes in the Surgical Cohort

The risk factors for adverse clinical outcomes after surgery were evaluated to inform identification of appropriate surgical candidates. These risk factors were analyzed in the surgical cohort using a logistic regression model. All variables identified as significant in the univariate analysis (Appendix A) were included in the multivariable model and were subsequently chosen by backward stepwise selection. Multicollinearity between covariates was tested using the variance inflation factor. The calibration of the multivariable model was assessed using the Hosmer–Lemeshow goodness-of-fit test. The estimates of the multivariable model were internally validated with a relative bias on 1000 bootstrapped samples.

### 2.6. Statistical Analysis 3: Evaluation of Clinical Risks from Additional Spinal Instrumentation in the Matched Surgical Cohort

Clinical risks from additional spinal instrumentation were thoroughly analyzed in the matched surgical cohort, comparing adverse clinical outcomes between the surgical groups based on whether additional instrumentation was conducted. Further 1:1 propensity score matching was performed to reduce potential selection bias. The matched covariables included age, body mass index (BMI), and other variables that showed intergroup differences from unmatched between-group comparisons. The rates of recurrence, major complications, and mortality were compared before and after matching (Appendix A). The clinical risks of early spinal instrumentation were further evaluated by multivariable analysis within the matched cohort using logistic and Cox regression models, adjusting for variables that were significantly different (*p* < 0.20) between groups after matching. All analyses were performed using SPSS software (version 25.0; IBM Corp., Armonk, NY, USA).

## 3. Results

### 3.1. Demographics and Completeness of Follow-Up 

A total of 439 older (aged ≥75 years) patients with PVO were finally included, with 194, 130, and 115 patients in the non-operative, non-instrumented, and instrumented groups, respectively (Figure 1). The median age was 81 years (range, 75–93 years), and 65% (285/439) were women. A minimum follow-up of one year was mandatory for inclusion. The rates of loss to follow-up were 19% (46 of 240), 12% (18/148 patients), and 7% (8/123 patients) in the non-operative, non-instrumented, and instrumented groups, respectively (Figure 1).

### 3.2. Clinical Characteristics of the Three Treatment Groups

The nonoperative group was older and had more severe comorbid medical conditions based on CCI score and lower bone mineral density at the spine than the surgical group (Table 1). On the other hand, the nonoperative group had lesser neurologic impairment by ASIA grade, less frequent abscess formation, and lower degree of infection based on Pola et al. (Table 2). Compared to the non-instrumented group, the instrumented group had less severe comorbid medical conditions, more frequent abscess formation, higher degree of infection by Pola et al., and higher levels of ESR and CRP. The precise causative organisms of the three treatment groups are presented in Appendix A. Intergroup differences were not observed in the occurrence of major medical events, recurrence rates, and 90-day and 1-year mortality rates.

Most patients underwent surgical treatment within 2 weeks after PVO diagnosis. Of those, there were 95 (73%) and 89 patients (77%) in the non-instrumented and instrumented groups, respectively (Appendix A). The posterior approach was the most commonly performed method, noted in 124 (95%) and 89 patients (77%) in the non-instrumented and instrumented groups, respectively.

### 3.3. Comparison of Treatment Outcome According to the Three Treatment Groups in the Whole Cohort

Upon 1-year follow-up, the recurrence-free survival rate was 89% (95% CI, 83–92) in the non-operative group, 85% (95% CI, 78–90) in the non-instrumented group, and 90% (95% CI, 84–95) in the instrumented group (Figure 2a). Upon 1-year follow-up, the death-free survival rate was 84% (95% CI, 78–89) in the non-operative group, 72% (95% CI, 63–79) in the non-instrumented group, and 86% (95% CI, 78–91) in the instrumented group (Figure 2b).

Based on multivariable analysis, the two surgical groups did not show an increased risk of recurrence compared to the nonoperative group (Table 3). However, the non-instrumented group showed significantly increased risk for 1-year mortality compared to the nonoperative group (OR: 2.67 [1.27–5.61], *p* = 0.010). However, the risk of overall mortality was not significantly different among the three groups.

### 3.4. Identification of Risk Factors for Recurrence or Mortality in the Surgical Cohort

The univariable analysis to identify risk factors for recurrence or 1-year mortality is presented in Appendix A. Multivariable analysis demonstrated that recurrence was associated with comorbidities, including liver cirrhosis (OR = 3.60), complicated diabetes (OR = 5.00), MRSA (OR = 4.35), and neurologic deficits (ASIA grades A, B, and C vs. grade E, OR = 4.28), whereas 1-year mortality was associated with CCI score (OR = 1.41 per point), liver cirrhosis (OR = 3.01), and neurologic deficits (ASIA grades A, B, and C vs. grade E, OR = 7.44) (Table 4). Age was not a significant risk factor for recurrence or mortality in this patient population. Multicollinearity among covariates in the multivariable model was low, and all variance inflation factors were <1.5. The Hosmer–Lemeshow goodness-of-fit test indicated good calibration (*p* = 0.396 for recurrence, *p* = 0.789 for 1-year mortality). After bootstrap adjustment, the relative biases of estimates in the multivariable model were low, ranging from −5.2 to 12.5% for recurrence (−5.2% for liver cirrhosis, 12.5% for complicated diabetes, 8% for ASIA grade, 7.5% for causative organism) and 1-year mortality (1.1% for CCI score per point, 4.1% for liver cirrhosis, and 9.1% for ASIA grade).

### 3.5. Evaluation of Clinical Risks of Additional Spinal Instrumentation in the Matched Surgical Cohort

The clinical risks of early instrumentation on recurrence and mortality were further evaluated using multivariable analysis within the matched surgical cohort (Table 5). Adjustments were made for age, ASIA grade, and number of infected vertebral bodies. After adjustment, there were no intergroup differences in recurrence (odds ratio [OR] = 1.29, *p* = 0.638), 90-d and 1-year mortality (OR = 1.43, *p* = 0.524; OR = 0.87, *p* = 0.764, respectively), or hazard ratios for overall mortality (OR = 1.12, *p* = 0.713).

## 4. Discussion

In the current study, which investigated the treatment outcomes of older patients with PVO, we tried to answer three research questions. First, we compared the surgical outcomes of older patients with PVO to those who underwent nonoperative treatment. After multivariable analysis, the two surgical groups did not show increased risk of recurrence compared to the nonoperative group (Table 3). There were also no significant differences in overall mortality among the three groups, although the non-instrumented group showed an increased risk for one-year mortality than the nonoperative group. Subsequently, we evaluated risk factors for adverse clinical outcomes after surgery to inform appropriate selection of surgical candidates. Multivariable analysis, using internal validation by bootstrapping, demonstrated that recurrence after surgical treatment for PVO was associated with comorbidities, including liver cirrhosis and complicated diabetes, MRSA, and severe neurologic deficit. On the other hand, 1-year mortality was associated with a higher CCI score, liver cirrhosis, and severe neurologic deficit (Table 4). Finally, the clinical risks from additional spinal instrumentation were thoroughly analyzed to evaluate its safety in older patients with PVO. Increased surgical invasiveness from additional spinal instrumentation did not increase the risk for recurrence and 90-day or 1-year mortality in older patients in whom surgical treatment was primarily performed via a posterior approach. 

The two major indicators for the treatment success of PVO are recurrence and mortality. Therefore, before proposing treatment guidelines based on our results from older patients with PVO, we need to compare the rates of recurrence and mortality, and their risk factors, to those of younger patients. Previous studies have consistently reported that clinical outcomes including postoperative recurrence and mortality for patients with PVO were closely associated with medical comorbidities, undrained abscesses, and causative organisms, including MRSA [7,10,17,18]. However, the risk factors for adverse clinical outcomes in our older patients aged ≥75 years with PVO differed slightly from those of previous studies that were based on younger populations. Recurrence and mortality in our patients were closely associated with neurologic deficit.

Our older patients with PVO showed high 90-d (9%, 39/439) and 1-year (20%, 86/439) mortality, which were considerably higher than those of younger patients, based on previous reports (1−13%) [7,8,9]. Mortality in older patients was predicted by the severity of comorbidities, as indicated by CCI score, rather than by specific diseases (Table 4), except for liver cirrhosis, which is known to be a strong, independent risk factor for mortality in PVO patients [17]. 

However, the recurrence rate in our older cohort was not higher (12%, 52/439 patients) than that in younger patients, based on previous reports (11–18%) [7,9,10,19]. The recurrence in these patients was predicted by specific diseases, such as liver cirrhosis and diabetes, rather than the severity of the comorbid disease (Table 4). MRSA remained a significant risk factor for recurrence in these patients. The favorable recurrence rate in older patients with PVO needs to be carefully interpreted. First, compared with studies based on younger patients with PVO, [8,10,11,17] the proportion of MRSA as a causative organism was low in older patients (20%, 88/439 patients), while the proportion of gram-negative bacterial infection was relatively high (37%, 161/439 patients). The lower proportion of MRSA in older patients with PVO might positively influence recurrence. However, attention should also be paid to older patients with gram-negative PVO. Generally, compared to patients with gram-positive bacteremia, those with gram-negative bacteremia have a poorer prognosis, with a more severe inflammatory response or more severe sepsis [20]. Second, a significant number of older patients with PVO expired within 1 year (20% in our cohort), and recurrence rates estimated from survivors could be biased. 

Furthermore, additional spinal instrumentation in older patients did not increase the risk of recurrence (Table 5). Recent studies have consistently reported insignificant differences in recurrence rates between instrumented and non-instrumented surgeries for PVO, [9,10] although instrumented surgery is generally performed in PVO patients with a severe degree of infection and mechanical instability. In those studies, more vigorous removal of abscesses during instrumented surgery, a known independent risk factor for PVO recurrence, [7,8,10] was suggested as a reason for similar recurrence rates between instrumented and non-instrumented surgical methods. Additionally, a relatively lower proportion of MRSA among causative organisms in older patients with PVO might have positively contributed to the similar recurrence rates.

Based on the study findings, we suggest the following treatment guidelines for surgical treatment in older patients with PVO. The primary objective for this patient population is to reduce mortality rates. Ten percent of the patients expire early (within 90 days), and early mortality occurs in hemodynamically unstable patients with PVO who have pre-existing, life-threatening, and severe comorbid conditions [17]. Regardless of the surgical strategy, it is difficult to prevent early mortality in these patients, and the timing of surgical treatment should be carefully decided after considering the hemodynamic stability of patients. If these patients survive during the early critical period, clinicians should focus on subsequent morbidity secondary to structural and neurologic injuries, resulting in later mortality. Late (90 days to 1 year) mortality in our patient cohort (11%, 47 of 439) was not only higher than that in younger patients, [7,8,9] but also higher than the corresponding 90-day mortality for the same age group (9%, 39 of 439). One-year mortality in the older population is closely associated with neurological deficits (Table 4). Therefore, we recommend immediate surgical treatment for hemodynamically stable older patients with PVO who have neurologic deficits or structural instability. Decompressive surgery, including sufficient removal of the abscess, should be performed to rapidly recover neurological deficits, and spinal instrumentation can be performed with reasonable clinical risk, if indicated. 

Although our study demonstrated comparable clinical risks in terms of recurrence or mortality among the different treatment groups, the results should be carefully interpreted considering possible biases and confounders. The three treatment groups were purposefully dissimilar at baseline and received different management not only in the selection of treatment methods but also in postoperative management and outpatient follow-up. Due to such unknown confounders, our results do not guarantee identical risks among the three treatment groups. Accordingly, we recommend that clinicians use the worst-case estimate from the Kaplan–Meier survival probability and its 95% confidence interval (Figure 2). Second, although the proportion of patients who were lost to follow-up was small, there may have been unrecorded recurrences or mortalities associated with infection. In particular, the effect of this transfer bias may be considerably increased in older patients. Third, our cohort patients were selected through a retrospective review of data during the extensive period between 2004 and 2020. Although surgical treatment was generally performed according to our institutional protocol, different surgical procedures, including debridement of additional bone and soft tissue, could be performed with regard to the surgeon’s discretion, based on intraoperative findings. This can be a very important confounder in the comparison of clinical outcomes in the surgical cohort. Fourth, our study did not assess actual clinical outcomes using a validated disability index or quality-of-life scale. Finally, the small sample size may have reduced the power of the study. 

In conclusion, the distribution of causative organisms and risk factors for adverse clinical outcomes in patients with PVO aged ≥75 years were different from those of previous studies that were based on younger populations. The proportion of MRSA as a causative organism was relatively low, and clinical outcomes, including recurrence and mortality, were closely associated with neurologic deficits. Moreover, increased surgical invasiveness, resulting from additional spinal instrumentation, did not increase the risk of recurrence or mortality in older patients whose surgical treatment was primarily performed via a posterior approach. Our findings suggest that surgical treatment for older patients with PVO should be focused on to improve neurologic deficits by sufficient and immediate removal of the abscess. Spinal instrumentation can be performed if indicated, with reasonable clinical risk.

## Figures and Tables

**Figure 1 jcm-10-05451-f001:**
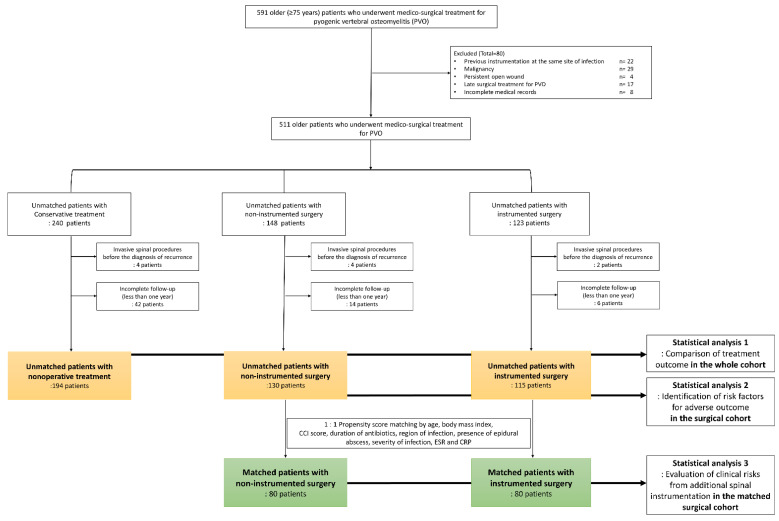
Patient enrollment.

**Figure 2 jcm-10-05451-f002:**
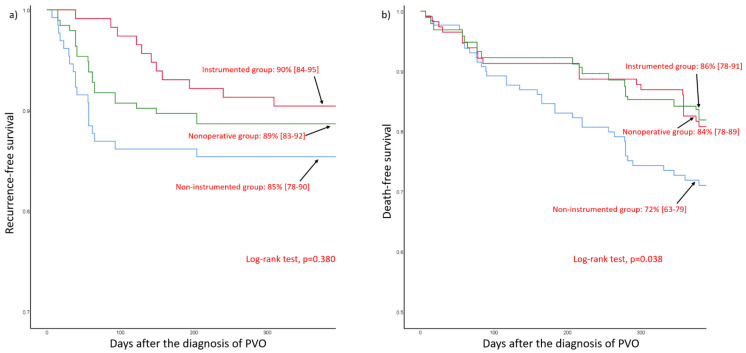
Kaplan–Meier survival curves. (**a**) Recurrence-free survival. (**b**) Death-free survival. Survival probabilities and corresponding 95% confidence intervals at 1-year follow-up are presented.

**Table 1 jcm-10-05451-t001:** Comparison of baseline patient characteristics among the three groups.

Variables	Category	Whole Cohort				
		Nonoperative Group	Surgical Group			*p*-Value ^2^
			Non-Instrumented Group	Instrumented Group	*p*-Value ^1^	
Number of patients		194	130	115		
Age	Median (Interquartile Ranges)	80 (80–85)	80 (79–83)	81 (79–82)	0.838	<0.001
	75 to 79 years	38 (20)	44 (34)	38 (33)	0.979	<0.001
	80 to 84 years	107 (55)	69 (53)	61 (53)		
	≥85 years	49 (25)	17 (13)	16 (14)		
Sex ratio (F:M)		126:68	86:44	73:42	0.661	0.991
BMI (kg/m^2^)		22.6 (20.9–24.6)	22.4 (21.2–24.5)	23.9 (21.0–26.3)	0.170	0.093
Charlson Comorbidity Index score	Median (Interquartile ranges)	3.0 (2.0–5.0)	3.0 (1.0–4.0)	2.0 (1.0–4.0)	0.041	0.006
	0 and 1	46 (24)	37 (28)	50 (43)	0.047	0.028
	2 and 3	79 (41)	49 (38)	36 (31)		
	over 4	69 (36)	44 (34)	29 (25)		
Comorbidities	Cerebrovascular Disease	31 (16)	27 (21)	26 (23)	0.727	0.135
	Myocardial Infarction	48 (25)	36 (28)	29 (25)	0.661	0.670
	Congestive Heart Failure	40 (21)	22 (17)	11 (10)	0.092	0.046
	Chronic Obstructive Pulmonary Disease	24 (12)	15 (12)	10 (9)	0.463	0.474
	Liver Cirrhosis	26 (13)	17 (13)	9 (8)	0.183	0.369
	End-Stage Renal Disease	56 (29)	41 (32)	27 (23)	0.160	0.797
	Diabetes				0.612	0.867
	Complicated ^3^	92 (47)	60 (46)	52 (45)		
	Uncomplicated	26 (13)	22 (17)	15 (13)		
Bone Mineral Density (g/cm^2^)	Spine	0.808 (0.721–0.919)	0.854 (0.732–1.010)	0.892 (0.777–0.981)	0.317	0.001
	Femur	0.639 (0.557–0.741)	0.645 (0.575–0.743)	0.700 (0.609–0.757)	0.101	0.082
American Spinal Injury Association Scale Grade	Grade A, B and C	9 (5)	18 (14)	25 (22)	0.118	<0.001
	Grade D	90 (46)	71 (55)	49 (43)		
	Grade E	95 (49)	41 (32)	41(36)		
Duration of Intravenous Antibiotics (Days)		52 (45–63)	44 (42–56)	52 (45–63)	<0.001	0.004
Hospital Stay (Days)		60 (53–72)	52 (47–59)	54 (47–63)	0.084	<0.001
Follow-Up Period		605 (387 -1132)	801 (478–1262)	1036 (597–1457)		

^1^ *p*-values for the difference between the non-instrumented and instrumented group. ^2^ *p*-values for the difference between the nonoperative and surgical group. ^3^ Complicated diabetes was defined as diabetes associated with damage to the end organs, including the cardiovascular system, kidneys, eyes, and nervous system, and it included peripheral neuropathy, nephropathy, retinopathy, and peripheral artery disease.

**Table 2 jcm-10-05451-t002:** Comparison of infection profiles among the three groups.

Variables	Category	Whole Cohort				
		Nonoperative Group	Surgical Group			*p*-Value ^2^
			Non-Instrumented Group	Instrumented Group	*p*-Value ^1^	
Primary Region ^3^	Cervical Spine	17 (9)	10 (8)	23 (20)	0.022	0.405
	Thoracic spine	77 (40)	52 (40)	48 (42)		
	Lumbosacrum	95 (49)	64 (49)	42 (37)		
	Multiple	5 (3)	4 (3)	2 (2)		
Presence of abscess	Epidural abscess	103 (53)	115 (88)	111 (97)	0.019	<0.001
	Posterior to epidural space	121 (62)	96 (74)	91 (79)	0.332	0.002
	Anterior to epidural space	89 (46)	81 (62)	66 (57)	0.433	0.003
Number of infected vertebral bodies	Within 2 vertebral bodies	51 (26)	95 (73)	77 (67)	0.296	0.418
	Over 3 vertebral bodies	143 (74)	35 (27)	38 (33)		
Severity of infection by Pola, et al.	Type A	4 (2)	2 (2)	0 (0)	0.015	<0.001
	Type B	86 (44)	13 (10)	4 (3)		
	Type C	104 (54)	115 (88)	111 (97)		
Causative organism	Staphylococcus aureus	78 (40)	52 (40)	47 (41)	0.586	0.978
	Methicillin resistant	40 (21)	24 (18)	24 (21)		
	methicillin sensitive	38 (20)	28 (22)	23 (20)		
	Other gram-positive bacteria	23 (12)	15 (12)	15 (13)		
	Gram-negative bacteria	70 (36)	46 (35)	45 (39)		
	Others or unidentified	23 (12)	17 (13)	8 (7)		
White blood cell (×10^9^/L)	Initial	12,399 (9926–15,556)	12,679 (10,105–15,436)	13,060 (10,330–16,799)	0.155	0.380
Erythrocyte sedimentation rate (esr, mm/h)	Initial	59 (44–70)	62 (47–71)	69 (53–75)	0.008	0.002
C-reactive protein (crp, mg/L)	Initial	71 (57–83)	69 (58–83)	75 (67–88)	0.036	0.180
Major medical events after treatment	At least one of the following complications	60 (31)	39 (30)	25 (22)	0.142	0.267
	Cardiac event	15 (8)	11 (8)	7 (6)		
	Respiratory complication	38 (20)	25 (19)	15 (13)		
	Cerebrovascular complication	5 (3)	4 (3)	4 (3)		
	pulmonary embolism	11 (6)	8 (6)	4 (3)		
Recurrence	occurrence	22 (11)	19 (15)	11 (10)	0.229	0.771
	Interval between initial diagnosis and recurrence (days)	55 (39–93)	39 (23–57)	142 (96–194)	<0.001	0.767
Mortality	90-day mortality	15 (8)	14 (11)	10 (9)	0.586	0.450
	1-year mortality	30 (15)	36 (28)	20 (17)	0.055	0.053

^1^ *p*-values for the difference between the non-instrumented and instrumented group. ^2^ *p*-values for the difference between the nonoperative and surgical group. ^3^ Multiple primary regions were defined as non-contiguous, separate foci of infection, involving at least two sites. If a single, continuous focus of infection exists involving two regions, this type of infection was classified based on the region with a more severe degree of involvement. Continuous data were presented by median and interquartile ranges.

**Table 3 jcm-10-05451-t003:** Comparison of treatment outcome according to the three treatment groups in the whole cohort: multivariable analysis.

Outcome	Category	Group	Odds Ratios/Hazard Ratios	95% Confidence Interval	*p*-Value
Recurrence ^1^		Non-instrumented group vs. nonoperative group	0.99	(0.45–2.20)	0.983
		Instrumented group vs. nonoperative group	0.75	(0.30–1.91)	0.547
Mortality	90-day mortality ^1^	Non-instrumented group vs. nonoperative group	1.88	(0.69–5.11)	0.218
		Instrumented group vs. nonoperative group	1.73	(0.58–5.20)	0.330
	1-year mortality ^1^	Non-instrumented group vs. nonoperative group	2.67	(1.27–5.61)	0.010
		Instrumented group vs. nonoperative group	1.35	(0.59–3.07)	0.477
	Overall mortality ^2^	Non-instrumented group vs. nonoperative group	1.67	(1.09–2.57)	0.019
		Instrumented group vs. nonoperative group	1.07	(0.65–1.75)	0.798

^1^ Multivariate logistic regression model. ^2^ Cox regression model. All adjustments were performed for age, Charlson comorbidity index score, bone mineral density at the spine, ASIA grade, duration of intravenous antibiotics, presence of the three types of abscesses, severity of infection, and erythrocyte sedimentation rate.

**Table 4 jcm-10-05451-t004:** Risk factors for recurrence and mortality in the surgical cohort: multivariable analysis.

Outcomes	Variables	Category	Multivariable (Backward)			Bootstrap-Adjusted	Bias
			Odds Ratio	95% CI	*p*-Value		
Recurrence	Comorbidities	Liver cirrhosis	3.60	(1.17–11.09)	0.026	3.79 (1.06–12.10)	−5.2
		Diabetes, complicated	5.00	(1.73–14.50)	0.003	4.38 (1.25–30.63)	12.5
	ASIA grade	Grade A, B and C vs. grade E	4.28	(1.36–13.52)	0.013	3.94 (1.01–25.30)	8
	Causative organism	MRSA vs. no MRSA	4.35	(1.67–11.35)	0.003	4.02 (1.10–19.87)	7.5
One-year mortality	CCI score	Per 1 point	1.41	(1.17–1.69)	<0.001	1.39 (1.13–1.79)	1.1
	Comorbidities	Liver cirrhosis	3.01	(1.04–8.74)	0.042	2.89 (0.93–13.05)	4.1
	ASIA grade	Grade A, B and C vs. grade E	7.44	(2.89–19.16)	<0.001	6.76 (2.34–31.63)	9.1

ASIA grade: American Spinal Injury Association Scale grade CCI score; Charlson comorbidity index score. All significant independent variables (*p* < 0.05) from the univariate analysis were initially included and subsequently chosen by backward stepwise selection in the multivariable model. Relative bias was estimated as the difference between the mean bootstrapped regression coefficient estimates and the mean parameter estimates of the multivariable model divided by the mean parameter estimates of the multivariable model.

**Table 5 jcm-10-05451-t005:** Clinical risks of additional spinal instrumentation in the matched surgical cohort: multivariable analysis.

Outcomes	Category	Group	Odds Ratios/Hazard Ratios	95% Confidence Interval	*p*-Value
Recurrence ^1^		Instrumented group vs. non-instrumented group	1.29	(0.45–3.73)	0.638
Mortality	90-day mortality ^1^	Instrumented group vs. non-instrumented group	1.43	(0.45–4.81)	0.524
	1-year mortality ^1^	Instrumented group vs. non-instrumented group	0.87	(0.36–2.11)	0.764
	Overall mortality ^2^	Instrumented group vs. non-instrumented group	1.12	(0.61–2.08)	0.713

^1^ Multivariate logistic regression model; ^2^ Cox regression model; All Adjustments were done for age, ASIA grade, and number of infected vertebral bodies.

## Data Availability

The data presented in this study are available on request from the corresponding author. The data are not publicly available due to ethical restriction.

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
