# Peer review of "Clinical Outcomes in Older Patients Aged over 75 Years Who Underwent Early Surgical Treatment for Pyogenic Vertebral Osteomyelitis"

_jcm, 2021, doi:10.3390/jcm10225451_

Round 1
Reviewer 1 Report
The authors investigate clinical outcomes in elderly patients who underwent early surgical treatment for vertebral osteodiscitis. They have an impressive number of patients over a long period of time (2004 to 2020).
However, with this amount of clinical information and the amount of long term follow up available, I would prefer to see a study that compares the patients who undergo early surgical treatment versus those that undergo a trial of antibiotics to see the differences in outcomes and the failure rates in the nonsurgical group.
The authors mention surgery was performed for mechanical instability, intractable pain, and/ or a neurological deficit. I think we can all agree that these would be reasons to perform surgery, however, many early cases of osteodiscitis do not have these characteristics and can certainly have successful outcomes without surgery.
For these reasons, I do not know that this manuscript will substantially add to the current understanding and treatment of spinal infections, and I would therefore recommend amending the study design.
Author Response
Our response to the first reviewer was added as a separate word file.

Reviewer 2 Report
Several comments,
- Introduction
- This section should clearly describe current knowledge about the topics that the authors want to discuss.
- Materials and Methods
- The distribution of patient is not in normal distribution. Therefore, median rather mean should be used.
- Advanced PYO, please clearly define this description.
- Early surgical treatment (≤6 weeks), why not use 2-week or even shorter duration as the definition?
- Early spinal instrumentation, what is the definition?
- The duration of antibiotic therapy, just including intravenous drugs?
- Recurrent signs and symptoms should be outlined.
- Remove most abscess? How to define, using radiological methods or judged objectively by the surgeons?
- The degree of surgical debridement should also be included in the method of propensity-matching score.
- For the description of mortality, 90-day or one-year is more appropriate than early or late.
- The tables and figures presented in the section of method should be re-placed in the section of results or supplementary.
- Results
- Table 1 (?), How to define complicated or uncomplicated DM?
- Table 1, the duration of antibiotic therapy includes only intravenous drugs or both intravenous and oral drugs?
- Table 2 (infection profile), the definition of mainly XX spine and multiple should be pointed out.
- Table 2, the details of micro-organism distribution should be described.
- In the sections titled as “Evaluation of clinical risks of early spinal instrumentation in the matched cohort” and “Risk factors associated with recurrence or mortality: multivariable analysis in the entire cohort with bootstrap validation”, these two parts can be integrated into one and the subtitle should be revised.
- Discussion
- Multivariable analysis of our study demonstrated that recurrence was associated with comorbidities, including liver cirrhosis and complicated diabetes, MRSA, and severe neurologic deficits with ASIA grades A, B or C, while 1-year mortality was associated with a higher CCI score, liver cirrhosis, and severe neurologic deficit. Unlike the findings from previous studies on younger patients with PVO, the clinical outcomes, including recurrence and mortality, of older patients with PVO who underwent surgical outcome were closely associated with neurologic….these descriptions are confused, please re-write
- In the other parts of discussion, it lacks flow and transition sentences between paragraphs. Therefore, it should be re-written.
- How to explain the difference in recurrence days between two groups?
- References
- Some of the references should be corrected. For example, references 14 and 16.
Overall, the whole article should be re-arranged (too many tables) and revised to be clearer and readable to the readers. The English grammar should be checked.
Author Response
Our response to the second reviewer was added as a separate word file.

Reviewer 3 Report
The authors of this manuscript analyze a very large retrospective series of elderly patients with vertebral osteomyelitis, with the aim of evaluating whether performing spinal instrumentation is a safe and efficient practice for infection control. The work is well done, the statistical analysis of the data is adequate and the conclusions are very relevant.
The main strength of the article is the large number of patients included and the rigorous presentation of the data and statistical analyses. The most important weaknesses would be: 1) the indication for the practice of spinal instrumentation is not well predefined and there could be clear patient selection biases in each of the therapeutic modalities; 2) a very extensive amount of information is provided that could be in the form of supplementary material.